# Heat on the Move: Contrasting Mobile and Fixed Insights into Temuco’s Urban Heat Islands

**DOI:** 10.3390/s25041251

**Published:** 2025-02-18

**Authors:** Aner Martinez-Soto, Michelle Vera-Fonseca, Pablo Valenzuela-Toledo, Aliwen Melillan-Raguileo, Matthew Shupler

**Affiliations:** 1Department of Civil Engineering, Faculty of Engineering and Science, Universidad de La Frontera, Temuco 4811230, Chile; m.vera06@ufromail.cl; 2Department of Computer Science, Faculty of Engineering and Science, Universidad de La Frontera, Temuco 4811230, Chile; pablo.valenzuela@ufrontera.cl (P.V.-T.); aliwen.melillan@ufrontera.cl (A.M.-R.); 3Department of Epidemiology, Harvard T.H. Chan School of Public Health, Boston, MA 02115, USA

**Keywords:** urban heat islands, mobile transects, fixed stations

## Abstract

This study evaluates the combined use of mobile transects and fixed stations to analyze atmospheric urban heat islands (UHIs’a) in Temuco, Chile. Data were collected using 23 fixed stations and 3 mobile transects traversing predefined city routes, capturing temperature records at one-minute intervals. Results revealed moderate correlations between methodologies (coefficients: 0.55–0.62) and average temperature differences of 0.72 °C to 1.6 °C, confirming their compatibility for integrated use. UHI intensities ranged from weak (0.5 °C) to extremely strong (13 °C), with the highest urban temperature (33.1 °C) observed in Zone Z-3, contrasting with 25.4 °C at the rural Maquehue station. Simulations and isothermal maps identified four UHI zones, highlighting the influence of impervious surfaces, traffic density, and limited vegetation on temperature distribution. Fluctuation plots revealed rapid cooling in vegetated areas and high heat retention in dense urban zones. These findings validate the methodologies for spatial and temporal UHI analysis and provide actionable insights for urban planning. Targeted interventions, such as increasing vegetation in high-risk zones, are recommended to mitigate extreme heat and enhance thermal comfort in urban areas.

## 1. Introduction

The expansion of urban centers and their growth into peripheral areas have significantly altered local weather patterns, primarily through changes in land cover and urban infrastructure [1]. This transformation has led to the replacement of natural surfaces, which dissipate radiative energy, with urban materials that absorb and retain heat, resulting in elevated temperatures in urban areas. This phenomenon, known as the urban heat island (UHI), is defined as the temperature difference between urban zones and their surrounding rural areas at a given time [2,3,4].

The UHI effect significantly increases energy demand for cooling in urban areas. For instance, in Tampico, Mexico, UHI effects have been shown to raise nighttime temperatures by up to 2.5 °C, reducing nighttime cooling and increasing indoor discomfort levels [5]. On a larger scale, energy consumption can rise by 10–25% during summer months due to the UHI effect, which also leads to higher emissions of greenhouse gases like carbon dioxide and nitrogen oxide, intensifying smog formation and contributing to global warming [6,7]. Such emissions are estimated to account for over 70% of urban air pollution during peak UHI events, particularly in regions with dense populations and limited vegetation [8]. Projections from the Intergovernmental Panel on Climate Change (IPCC) indicate that global temperatures may rise by 1.4 °C to 5.8 °C by 2100, compared to 1990 levels [9,10]. Extreme cases, such as Kuwait City, highlight the severity of these issues, with annual temperature increases of 0.3 °C to 0.8 °C and frequent occurrences of temperatures exceeding 50 °C [11].

Heatwaves, intensified by global warming, further amplify the UHI effect, impacting human health and comfort. For example, the 2003 European heatwave caused an estimated 70,000 deaths, while similar events in Russia (2010) and India (2015) led to over 20,500 fatalities, with UHI effects compounding the risks [4,12]. UHI intensities have been observed year-round, during both day and night. In Iași, Romania, nighttime UHI intensities range from 3.0 °C to 4.0 °C, while Mexico City experiences daytime intensities of 3.0 °C to 5.0 °C during the rainy season [13,14]. In Phoenix, Arizona, UHI intensities range from 0.8 °C to 5.4 °C depending on seasonal conditions [15,16]. These findings emphasize the global prevalence of UHIs, regardless of geographic location.

Several methodologies have been developed to quantify UHI intensity, including mobile transects, fixed weather stations, and satellite imagery. Mobile transects, used for over 40 years, provide spatially detailed data but are limited to capturing static moments. For instance, Martínez (2014) measured UHI intensities of up to 4.5 °C in Alicante, Spain, but the method failed to depict diurnal variability [17]. Fixed weather stations offer continuous data, as demonstrated in Athens, Greece, where UHI intensities reached 5 °C during the summer [14,17]. However, such networks are costly to maintain and often insufficient for spatial coverage [18]. Satellite imaging provides extensive spatial coverage, as seen in Johannesburg, where UHI intensities of 2.0–3.5 °C were observed. Yet, this method primarily measures surface temperatures and may not correlate with air temperatures, leading to potential errors of up to 1.68 °C, as seen in Apatity, Russia [19,20]. Emerging techniques and frameworks have furthered UHI analysis. Local Climate Zones (LCZs), a classification system for urban areas based on land use and morphology, improve UHI mapping by considering factors such as building density, vegetation, and surface type [16]. In Xi’an, China, LCZ-based studies revealed that UHI intensities in compact high-rise zones exceeded those in open low-rise zones by up to 5.6 °C, with shielding effects from tall buildings also altering pedestrian-level temperatures [21]. Similarly, integrating crowdsourced data, such as temperature sensors embedded in personal vehicles, enables high-resolution UHI mapping, revealing localized cooling effects from parks and green spaces. For instance, in Rennes and Dijon, France, green areas reduced temperatures by up to 3.5 °C [22]. Despite these advancements, the simultaneous validation and comparison of methodologies remain underexplored. For example, in Utrecht, combining fixed station and mobile transect data reduced measurement discrepancies by over 15%, demonstrating the potential for integrated approaches [23]. However, such efforts are rare in under-represented regions like Chile, where most UHI studies focus on central areas, using either mobile transects or satellite data [1,24,25]. Southern cities, such as Temuco, remain largely unexamined despite their unique climatic and urban characteristics, including high annual precipitation exceeding 1200 mm and heterogeneous land cover [26,27].

This study aims to measure and analyze UHI intensity in Temuco, Chile, through the combined application of fixed station and mobile transect methods. By validating the integration of these approaches and georeferencing UHI zones, this research provides a robust framework for understanding UHI dynamics in diverse climatic regions and supports urban planning strategies to mitigate its impacts on quality of life.

## 2. Methodology

### 2.1. Study Area: Temuco

Temuco, located at latitude 38°44′00″ S and longitude 72°35′00″ W, spans an area of 464 km^2^. The city is geographically defined by the Ñielol and Conunhueno hills, with altitudes of 335 m and 360 m above sea level, respectively. The Cautín River traverses the city, flowing northeast to southwest and contributing to the region’s microclimatic conditions [28]. These geographic features contribute to the region’s distinct microclimatic conditions, including localized variations in temperature and humidity. Temuco has experienced steady population growth over recent decades. Between 2002 and 2015, its population increased from 227,086 to 275,617 inhabitants [29,30]. This urban expansion has led to the construction of extensive residential neighborhoods, replacing vegetation with impervious surfaces such as concrete and asphalt. These materials have higher heat retention capacities, which significantly contribute to the urban heat island (UHI) phenomenon, resulting in increased urban temperatures [31]. Studies in similar urbanizing regions suggest that such land cover changes can amplify daytime temperatures by 2–3 °C while reducing nighttime cooling [32]. Temuco’s climate is classified as Csb under the Köppen climate classification, characterized by cold winters and dry summers, with mean annual temperatures exceeding 10 °C. Typical maximum and minimum temperatures range from 23 °C to 1 °C, respectively [33]. However, recent trends indicate a departure from historical climatic norms. Temperature data from the Maquehue weather station (38°44′00″ S, 72°35′00″ W) reveal that 6.65% of days between 2000 and 2019 recorded maximum temperatures above 30 °C, compared to only 3.2% of days during the 1980–2000 period. This doubling of extreme heat events underscores a clear warming trend. A record high of 40.2 °C was observed on 15 February 2019, at 3:00 PM, highlighting the intensifying thermal extremes. Seasonal analysis shows February accounts for approximately 50% of the highest temperatures, followed by January (43.75%) and March (6.25%).

The region’s warming trend aligns with broader climatic shifts observed across southern Chile. Studies suggest that regional warming, exacerbated by global climate change, is particularly pronounced in urbanized areas like Temuco, where vegetation loss and high urban density intensify heat retention [34]. Moreover, UHIs in Mediterranean climates, such as Temuco, are particularly severe during summer, with differences of up to 9 °C between urban and rural areas observed in similar Chilean cities [35]. Temuco’s rapid population growth, urban expansion, and distinct climatic and geographic features make it a relevant study area for examining UHI effects. This study aims to quantify UHI intensity in the city by integrating fixed weather station data with mobile transects, providing a comprehensive understanding of the phenomenon and its implications for urban planning and public health.

### 2.2. Procedure to Contrast Heat Island Location Methodologies

Urban heat islands (UHIs) are classified into two distinct types: surface urban heat islands (UHIs’s) and atmospheric urban heat islands (UHIs’a) [36]. Surface UHIs refer to the thermal differences between artificial and natural surfaces, while atmospheric UHIs refer to the air temperature differences observed between various urban zones [36]. Each type of UHI requires specific methodologies to determine its intensity. Satellite image analysis is typically employed to assess UHIs’s, as it captures surface temperatures over large spatial areas. In contrast, UHIs’a, which focus on air temperatures, are evaluated using methods such as mobile transects or fixed weather stations [37]. In this study, the focus was on UHIs’a, as they provide a direct understanding of temperature impacts on human comfort and energy use. Data for UHI’a analysis were collected using two complementary methodologies: mobile transects and fixed stations. Both approaches rely on air temperature measurements but differ in their spatial and temporal resolutions. Mobile transects, conducted with high-resolution sensors mounted on moving vehicles, offer detailed spatial data across urban and rural zones. Fixed stations provide continuous temporal data at specific locations, allowing long-term trends to be observed.

These complementary datasets enable a robust comparison of UHI’a intensity and distribution. The datasets generated by the two methodologies were contrasted using three key statistical measures: Pearson’s correlation coefficient, average temperature difference, and standard deviation. Pearson’s correlation coefficient was used to assess the degree of similarity between the two datasets, providing a quantitative measure of methodological consistency (see Table 1 for interpretation). Correlation values ranging from 0.75 to 1.0 indicate strong alignment, reflecting a high degree of methodological reliability [32]. In addition to correlation, average temperature differences and standard deviations were calculated to evaluate methodological precision and variability. The choice of Pearson’s correlation coefficient is supported by studies demonstrating its effectiveness in comparing urban and rural temperature profiles. For example, Moreno et al. (2023) highlighted its utility in understanding urban–rural thermal contrasts and vegetation’s mitigating effects [32]. Similarly, Verichev et al. (2023) emphasized that statistical metrics like standard deviation are critical for interpreting variations in cooling degree days (CDDs) across urban landscapes, further validating its inclusion in this study [38].

Furthermore, methodological comparisons in similar studies have demonstrated the unique strengths and limitations of mobile transects and fixed stations. Mobile transects are particularly valuable for mapping spatial gradients of UHI intensity, capturing urban-to-rural transitions that are often missed by fixed stations [34]. On the other hand, fixed stations provide continuous, high-frequency data, making them ideal for temporal analysis of diurnal and seasonal UHI patterns. By integrating these methods, this study achieves a more comprehensive analysis of UHI’a intensity in Temuco.

#### 2.2.1. Mobile Transect Method

The mobile transect method involves collecting air temperature data by traversing predefined routes through the city, capturing localized temperature variations in urban and surrounding areas [40]. In this study, three mobile transects were conducted simultaneously using three vehicles equipped with HOBO Pendant MX2201 and MX2202 sensors (HOBO®, Bourne, MA, USA). These sensors are responsive to thermal variations, with a measurement range of −20 °C to 70 °C and an accuracy of ±0.5 °C. Their specifications ensure precise temperature data collection, necessary for analyzing the urban heat island (UHI) phenomenon.

Before data collection, the three sensors were calibrated to ensure consistent measurements. Calibration involved recording air temperatures under conditions with direct solar radiation and minimal shading to test sensor accuracy. The calibration process verified that the maximum variation among sensors was ±0.15 °C, ensuring reliability during the study. The sensors were mounted on the passenger-side windows of the vehicles to avoid interference from engine heat and urban obstructions. The vehicles traveled at an average speed of 30 km/h, which allowed for the collection of detailed temperature gradients while maintaining consistency across all routes. Each data point was georeferenced using GPS, and the data were downloaded via Bluetooth using the HOBOmobile platform, available on Android, iOS, and Windows. Measurements were conducted between 20:00 and 20:30 to minimize the influence of direct solar radiation. At this time, urban infrastructure begins releasing heat absorbed during the day, offering a representation of UHI effects. This timing aligns with standard practices for UHI analysis and ensures comparability with previous studies.

The design of the transect routes was informed by previous UHI analyses conducted in Temuco on 27 November 2017 and 19 January 2018, when the highest annual temperatures of 28.5 °C and 31.4 °C were recorded, respectively. The data from these dates identified four key UHI zones in the city: Z-1, Z-2, Z-3, and Z-4 (Figure 1). These zones were characterized by distinct environmental and urban features. Z-1 (Coord.: −38.709179, −72.554418) and Z-3 (Coord.: −38.742428, −72.643479) showed limited vegetation within urbanized areas compared to surrounding green spaces, leading to significant heat retention during the day. Conversely, Z-2 (Coord.: −38.753900, −72.624335) and Z-4 (Coord.: −38.726355, −72.619119) experienced heavy traffic congestion, contributing to high anthropogenic heat emissions.

Based on these findings, three simultaneous routes were defined to maximize spatial coverage and accurately represent the UHI phenomenon in Temuco (Figure 2). Transect 1 (red) covered a total distance of 10.50 km, traversing the city from north to south along Avenida Caupolicán and passing through Z-1 and Z-2. Transect 2 (magenta) spanned 10.35 km from the western sector of the city to the northeast, covering Z-3 and Z-4. Transect 3 (green) extended 10.73 km, initially crossing the city from north to south before following the Cautín River in an east–west direction through Z-2. The overlapping endpoints of Transects 2 and 3 were designed to analyze temperature reduction rates across the period of measurement.

To confirm that the chosen routes represented the thermal profile of the UHI, a prior simulation of the UHI phenomenon was performed using isothermal maps derived from fixed station data (discussed in Section 2.3). This simulation supported the selection of routes and ensured that the mobile transects captured the spatial distribution of temperature variations across the city.

#### 2.2.2. Fixed Station Method

For the fixed station method, this study utilized temperature data collected by the National Monitoring Network (ReNaM). This network consists of 23 intelligent sensors (Netatmo), represented as black dots in Figure 2. These sensors are installed in private homes distributed across various zones in Temuco, providing spatial coverage for monitoring urban heat island (UHI) intensity.

The Netatmo weather stations comprise two devices (interior and exterior) made of UV-resistant aluminum. The exterior sensors operate within a measurement range of −40 °C to 65 °C with an accuracy of ±0.3 °C. To ensure data reliability, the outdoor sensors are protected from direct solar radiation and precipitation, minimizing potential sources of error. Temperature data are captured at 30-min intervals, following a predefined schedule (e.g., 8:00, 8:30, 9:00) to standardize the time of measurements and reduce temporal variability. These sensors were calibrated and validated by the Ministry of Housing and Urban Development (MINVU) in conjunction with the Chile Foundation, ensuring alignment with national standards.

To ensure consistency between datasets collected via the fixed stations and the mobile transects, the temperature data from the HOBO and Netatmo sensors were compared prior to analysis. This comparison showed an average variation of 0.48 °C between the two devices, indicating that the potential influence of differing sensors on temperature measurements was minimal.

### 2.3. Methodologies for UHI Simulation and Location in Temuco

This study utilized a structured methodological framework to simulate and locate urban heat islands (UHIs) in Temuco. The framework consists of three main stages: analysis and data preprocessing, data formatting and sampling, and image generation using processed data. These steps were implemented to ensure an accurate representation of the UHI phenomenon and to enable visualization through heat island maps and fluctuation plots [41].

The first stage involved analyzing temperature data collected on 4 December 2019, when the Maquehue station recorded a maximum temperature of 25.4 °C, one of the highest temperatures observed that month. Data preprocessing included correcting records with erroneous numerical formats, filtering out dates without associated data, and generating a standardized CSV file containing the parameters Date–Time, Station Identifier, Temperature (°C), Latitude, and Longitude (Table 2). This processed file served as the input for the image generation process.

Two types of visualizations were created to analyze UHI characteristics: heat island maps and fluctuation plots. The heat island maps depict the UHI phenomenon at specific times by plotting isotherms based on recorded temperature data. These isotherms were constructed using Delaunay’s triangulation, which connects data points into non-overlapping triangles and ensures a structured spatial interpolation of temperature distributions [42]. Each isotherm was represented in a distinct color, with red indicating high temperatures and blue representing cooler zones [42]. Fluctuation plots, on the other hand, illustrate 24-h temperature variations recorded at the stations with the highest and lowest temperatures during the study period. Data from the official Maquehue weather station were also included to provide an external reference point outside the urban area.

The intensity of UHIs was calculated for each time period using the following equation:$$UHI\;Intensity=T_{U,max}-T_{R,min}$$
where

$T_{U,max}$: Maximum urban temperature recorded within the city.$T_{R,min}$: Minimum temperature recorded in the surrounding rural areas, serving as a reference for non-urban conditions

UHI intensities were classified according to Table 3, which categorizes the effects into five levels: weak, moderate, strong, very strong, and extremely strong. This classification provides a standardized interpretation of temperature differentials observed during the study. To reconcile the differing temporal resolutions of fixed station and mobile transect data, the fixed station data (recorded at 30-min intervals) were interpolated to match the one-minute frequency of the mobile transects. Linear interpolation was used to estimate intermediate temperature values, ensuring consistency and comparability across datasets.

The methodology integrates robust data preprocessing, standardized classification frameworks, and advanced spatial analysis techniques to visualize UHI phenomena. To ensure consistency and comparability between the fixed station and mobile transect methodologies, all temperature measurements were conducted at a standardized height of 1.5 m above the ground. This height was chosen to align with common practices in UHI studies and minimize variability introduced by differing sensor elevations [44,45,46,47]. Maintaining the same height for both methodologies reduces the potential influence of ground-level heat, which can significantly elevate temperatures closer to the surface [48]. This approach ensures that the recorded temperatures reflect ambient air conditions rather than localized ground heat, thereby enhancing the reliability of the data for UHI analysis. By standardizing sensor height, the study provides a robust basis for comparing spatial and temporal variations in temperature between methodologies. Despite these strengths, potential challenges such as sensor placement biases and interpolation uncertainties were mitigated through sensor calibration, validation, and data cross-referencing to enhance reliability and accuracy.

## 3. Results

### 3.1. Temperature Patterns and Spatial Variability Across Transects

Figure 3, Figure 4 and Figure 5 illustrate a comparison of the temperature records obtained using the mobile transect and fixed station methodologies. These records were compared to determine how close the values are that these methodologies provide and to be able to validate their use separately or combined.

Transect 1 covered the route represented in red in Figure 2 of 10.5 km, where temperature data were captured every one minute. The route took thirty minutes, where thirty points were captured along the route. Figure 3 provides the graphical representation of the temperatures recorded by the transect contrasted with the temperatures obtained by isothermal maps given by the fixed station methodology. It is observed that both methodologies yield similar temperature records. From record N° 1 to record N° 12, there is an increase in temperature, after which there is a decrease until record N° 15. The similarity between the records obtained with the two methodologies is confirmed with the correlation coefficient of 0.55.

The greatest maximum temperature difference between the two methodologies at the same point reached 1.76 °C, located at the moment at which the city is entered, where the change in temperature is considerable. However, the average temperature difference captured by the two methodologies along the entire route was 0.72 °C, which implies that the classification of the UHI phenomenon is not altered by the points that have the greatest temperature difference. The standard deviation between the two temperature datasets is 0.5. This value is very close to the accuracy of the sensor, which implies that the dispersion of the data is not significant.

Transect 2 performed the route represented by magenta in Figure 2, which took 26–36 min. The circuit covered 10.35 km, where 36 temperature records were captured. It might be considered that, unlike Transect 1, the route ran through sectors with high traffic congestion, which acquired a larger amount of temperature data in a shorter distance. In Figure 4, it is possible to visualize with the fixed station methodology that there are four zones in the city with high temperatures. These zones are not clearly reflected in the mobile transect methodology. However, it is noted that the intensity of both temperature records decreases until record N° 29, when they begin to increase their intensity again. The similarity between the values is confirmed with the correlation coefficient of 0.61, a value classified as mean correlation.

The greatest maximum temperature difference captured by both methodologies at the same point was 2.91 °C, whereas the average temperature difference captured by both methodologies at the same point was 1.6 °C, a lower value than the ranges for which the UHI is defined, which does not alter its classification. The standard deviation that yields this temperature dataset is 0.65. This value indicates that despite analyzing the intensity of the phenomenon with the most distant record, its classification did not change.

Transect 3 performed the route represented in green in Figure 2 of 10.73 km, where data were captured for 30 min, creating a record of 30 points. Although the distance was greater than the other transects, the sector encompassed has few traffic lights, which makes the traffic steady and flow better. Figure 5 shows that the similarity in the two records is greater than Transects 1 and 2. With both methodologies it is observed that the journey goes between zones in the city that have a lower temperature to zones with a higher temperature. Figure 5 shows that both temperature intensities of the zones covered begin to decrease until record N° 8, the point where the temperatures in the zones begin to increase. This similarity that the records have in the increase and decrease in their intensity in similar ranges is confirmed by the correlation coefficient of 0.62. This value is classified as mean correlation.

The greatest maximum temperature difference captured was 2.7 °C, whereas the average temperature difference captured by both methodologies at the same point was 1.2 °C, a lower value than the ranges for which the UHI intensity is defined that does not alter its classification. The standard deviation obtained from this temperature dataset is 0.91, a range in which the data tend to disperse.

### 3.2. Location of Heat Islands in Temuco

Figure 6 shows the isothermal map (left) and fluctuation plot (right) generated at the time and in the sector where the city reaches its maximum temperature. The maximum temperature reached in Temuco is visualized on the isothermal map in Zone 3 (Coord.: −38.740785, −72.645474) at 14:00 h with 33.1 °C. The minimum temperature reached in the city at the same time was located in Zone 2 (Coord.: −38.745916, −72.633190), with 29.7 °C. The intensity of the UHI phenomenon was 3.4 °C, classified as a moderate UHI. However, when calculating the intensity of the phenomenon on the basis of the temperature of the Maquehue station at 14:00 h (20.1 °C), 13 °C of difference is obtained between the urban and rural sectors, which classifies the UHI as extremely strong. The fluctuation plot (Figure 6, right) also shows that the maximum temperature in the city (Stn. 288) increases and decreases in short periods of time. This may be due to the fact that in the zone where the temperature peak is reached, there is abundant vegetation and most of the infrastructures are isolated houses, which allow for rapid air circulation and therefore reduce the temperature in the sector. Additionally, this area, located in the suburban outskirts of the city, has historically maintained a high percentage of green spaces and vegetation due to its prior classification as a rural zone. Recent urban development has introduced residential buildings, yet the general layout continues to support efficient air flow due to its flat terrain and dispersed infrastructure. These characteristics collectively contribute to the localized reduction in temperature, emphasizing the role of vegetation and urban design in mitigating UHI effects in this region.

In Figure 6, it is observed that in addition to the zone with the maximum temperature (Z-3), there are two sectors that tend to have a higher temperature than the rest of the city (Z-1 and Z-4).

Figure 7 (left) shows the 24-h temperature fluctuation at station 247 in Zone 4 (Coord.:−38.733095, −72.6031539), where a temperature of 30.2 °C was recorded at 14:00 h. When calculating the temperature difference in the sector compared to the temperature at the Maquehue station at the same time, a UHI intensity of 10.1 °C was obtained, a value classified as an extremely strong UHI. However, when calculating the temperature difference that exists between city sectors, an intensity of 0.5 °C was obtained, classifying it as a weak UHI. Figure 7 (right) represents the temperatures at station 256 in Zone 1 (Coord.: −38.697133, −72.535842), which reached a temperature of 31.8 °C at 14:00 h. When determining the intensity of the UHI phenomenon, there was a difference of 11.7 °C, classifying it as an extremely strong UHI. By linking the minimum temperature found in the city to the temperature in the sector at the same time, a temperature difference of 2.1 °C was obtained, classifying it as a moderate UHI. When comparing any point of the city with an external point, intensities over 10 °C were reached. This would mean that the UHI phenomenon would influence the microclimate in every sector in the city.

## 4. Discussion

The 3D representation of temperature variations across mobile transects in Temuco (Figure 8) highlights significant spatial heterogeneity, particularly at the intersections of transects. These variations underscore the localized thermal dynamics shaped by urban morphology and anthropogenic factors. For example, at the downtown intersection of Mobile Transects 1 and 2, temperatures differ by approximately 5 °C, with Transect 1 recording near 25 °C and Transect 2 around 20 °C. Similarly, in Zone 2 (Z-2), Transect 1 registers temperatures near 23 °C, contrasting with Transect 3 at approximately 18 °C. These findings corroborate research demonstrating significant microclimatic variability even within short distances, influenced by urban structures and heat retention properties [49,50].

Such variations are consistent with studies emphasizing the critical role of land use and cover changes in shaping urban microclimates [51,52]. Urban centers, dominated by impervious surfaces and high-density constructions, intensify the urban heat island (UHI) effect through reduced evapotranspiration and increased thermal inertia [53,54]. Conversely, areas with greater vegetation cover demonstrate cooler temperatures, underscoring the role of green spaces in mitigating UHI effects [51,55]. The observed variations at transect intersections further reflect the methodological challenges inherent in mobile monitoring campaigns, as highlighted in other studies. For instance, differences in timing, measurement resolution, and urban characteristics can contribute to discrepancies in recorded temperatures [50,56]. This underscores the importance of integrating mobile and fixed monitoring systems to enhance spatial and temporal data resolution [52,54].

From an urban planning perspective, these findings underline the need for targeted interventions. Strategies such as increasing vegetation cover, optimizing urban design, and enhancing surface albedo are pivotal for improving thermal comfort [53,55]. For instance, compact high-rise zones have been shown to exhibit the highest temperature disparities due to reduced vegetation and extensive impervious surfaces [21]. Similarly, in Temuco, high-density zones have been identified as hotspots with maximum UHI intensities reaching up to 13 °C. The integration of green infrastructure, particularly in high-risk zones like Z-1 and Z-3, could mitigate extreme heat events, aligning with global best practices while addressing local climatic challenges.

The integration of diverse methodological approaches has significantly advanced UHI research. Mobile transects provide granular spatial data, complementing fixed station records and remote sensing observations. Studies utilizing low-cost sensors on mobile vehicles have demonstrated the feasibility of mapping urban temperature gradients while maintaining cost efficiency [52].

Further, the differences between transects illustrate the complexity of UHI phenomena, which are influenced not only by surface materials and vegetation but also by meteorological conditions and human activity patterns. Studies have noted that such variability complicates the development of universal mitigation strategies and highlights the necessity for localized approaches tailored to specific urban contexts [49,50,52,57]. Incorporating Local Climate Zone (LCZ) classifications has further standardized UHI comparisons across urban morphologies, revealing the influence of building density and vegetation on cooling demands [38]. These approaches are reflected in Temuco, where thoughtfully designed transect routes have enhanced the understanding of localized UHI dynamics.

Mitigation strategies such as urban greening have demonstrated measurable impacts on UHI reduction. Research indicates that increasing vegetation cover can decrease summer daytime UHI intensities by up to 1.5 K, contributing to both thermal regulation and urban sustainability [58,59]. In Temuco, integrating these measures could improve thermal comfort in vulnerable zones while promoting sustainable urban development. However, localized adaptations are necessary, as the cooling potential of vegetation varies with species and urban context.

The use of thresholds derived from existing studies, such as those presented in Table 3, provides a general framework for classifying UHI intensity. However, these thresholds are not tailored to the specific climatic and urban conditions of the study area. For instance, the thresholds in this study are adapted from reference [43], which focuses on Spain and its surrounding regions. While they provide a useful benchmark, applying them to a city like Temuco, with distinct geographical and climatic characteristics, may introduce inaccuracies. Temuco’s high annual precipitation, heterogeneous urban layout, and mix of dense and vegetated zones suggest the need for locally derived thresholds to enhance classification accuracy. Future research should prioritize the development of region-specific thresholds to better reflect local temperature patterns and urban dynamics, particularly in under-represented areas like southern Chile. This limitation has been acknowledged and highlights the importance of continuous methodological refinement in UHI studies.

The selection of the 20:00–21:00 timeframe for temperature measurements aligns with established best practices for UHI analysis, as this period minimizes the influence of solar radiation and allows for stabilized temperature conditions [60,61,62]. This timeframe is particularly critical in ensuring reliable comparisons between urban and rural zones, as it mitigates variability caused by daytime solar heating or nighttime cooling dynamics [63,64]. While the observed consistency between methodologies during this timeframe supports the validity of the approach, measurements at other times of the day may introduce greater variability, potentially affecting the comparability and classification of UHI intensities [65]. This limitation highlights the importance of selecting appropriate measurement windows tailored to specific study objectives and regional climatic conditions. Future studies could explore how diurnal variability influences UHI patterns to provide a more comprehensive understanding of temporal dynamics.

Anthropogenic waste heat plays a significant role in shaping UHI dynamics, particularly in urban areas with high traffic density and industrial activity [66]. Fixed stations, often located near infrastructure, are prone to capturing localized emissions from sources such as vehicle exhaust, industrial operations, and building heat dissipation, which can introduce biases in temperature measurements [67,68]. Mobile transects, while offering broader spatial coverage, are susceptible to transient influences, such as momentary emissions from passing vehicles or varying industrial activity along the route [69,70]. These factors can create variability in the data, complicating direct comparisons. This study highlights the need to consider such anthropogenic contributions when interpreting UHI measurements and suggests integrating both methodologies to better capture the spatial and temporal distributions of waste heat influences.

Finally, future research should address the socio-economic dimensions of UHI mitigation, considering that heat impacts disproportionately affect vulnerable populations. Integrating equity frameworks into UHI studies will enhance the relevance of findings across diverse urban settings and guide inclusive urban planning in Temuco and beyond. Advanced remote sensing and predictive modeling should also be explored to refine UHI assessments and support evidence-based policy decisions [34].

## 5. Conclusions

This study examined the combined use of mobile transects and fixed stations to analyze atmospheric urban heat islands (UHIs’a) in Temuco, Chile. The methodologies were validated through a comparative analysis, demonstrating moderate compatibility with correlation coefficients of 0.55, 0.61, and 0.62 for three separate transect routes. Average temperature differences between the two methods were 0.72 °C, 1.6 °C, and 1.2 °C, respectively, while standard deviations ranged from 0.5 to 0.91 °C. These results confirm that both methodologies can be reliably used, either independently or complementarily, to assess UHI intensity.

Mobile transects enabled the detailed mapping of temperature gradients across urban zones, while fixed stations provided continuous temporal data and a broader perspective on UHI behavior. For example, fixed station data highlighted significant diurnal variations, with maximum temperatures in urban zones reaching 33.1 °C at 14:00, compared to 25.4 °C at the rural Maquehue station. These temperature differences translate to UHI intensities classified as weak to moderate within the city (0.5 °C to 3.4 °C). However, when compared to rural areas, intensities ranged from 10.1 °C to 13 °C, categorizing Temuco’s UHI as extremely strong. Such intensities represent one of the highest values reported in global UHI studies and pose a significant risk to public health, particularly during heatwave conditions.

The study identified four zones within Temuco exhibiting pronounced UHI characteristics, influenced by high traffic congestion and reduced vegetation cover. For instance, Z-1 recorded some of the highest temperatures due to impervious surfaces and limited green infrastructure, while Z-4 experienced significant heat retention from heavy traffic. These findings align with global trends, where compact high-density urban areas consistently display elevated UHI intensities.

The fixed station methodology further facilitated the creation of isothermal maps and fluctuation plots, which revealed patterns such as higher afternoon temperatures and rapid cooling at night in zones with abundant vegetation. These maps not only validated the mobile transect routes but also provided critical data for visualizing UHI dynamics over 24-h cycles. In Temuco, the strongest UHI intensities occurred during the afternoon, deviating from typical evening peaks observed in many other cities, emphasizing the importance of localized studies to account for unique climatic and urban factors.

By comparing and validating two complementary methodologies, the study establishes a robust framework for UHI analysis in diverse urban contexts. The integration of mobile transects and fixed stations provides a cost-effective and scalable approach for high-resolution spatial and temporal assessments. Moreover, the findings underscore the importance of addressing urban heat as part of broader urban planning and public health strategies. In Temuco, targeted interventions such as increasing vegetation cover in high-risk zones like Z-1 and Z-4 could mitigate extreme temperatures and improve thermal comfort.

The comparison between fixed station and mobile transect methodologies highlights the complementary strengths and limitations of each approach. Fixed stations offer continuous temporal data, allowing for the detailed analysis of diurnal and seasonal temperature variations under controlled conditions [71,72]. However, their limited spatial coverage may overlook localized temperature gradients within urban areas [73]. Conversely, mobile transects provide high spatial resolution, capturing urban-to-rural transitions and localized thermal dynamics that fixed stations may miss [74,75]. Despite these strengths, mobile measurements are subject to greater operational variability, including the need for frequent calibration and transient environmental influences [76]. The integration of these methodologies addresses these individual limitations, offering a more comprehensive framework for UHI analysis.

Future research should explore the socio-economic implications of UHIs, including their impact on energy consumption, public health, and social equity. Expanding this framework to other cities with similar climatic and urban characteristics would enhance the generalizability of the findings. Additionally, integrating advanced technologies such as remote sensing and machine learning could improve the accuracy and predictive capacity of UHI assessments. These efforts will be critical for developing adaptive strategies to mitigate UHI effects and enhance urban resilience in the face of climate change.

## Figures and Tables

**Figure 1 sensors-25-01251-f001:**
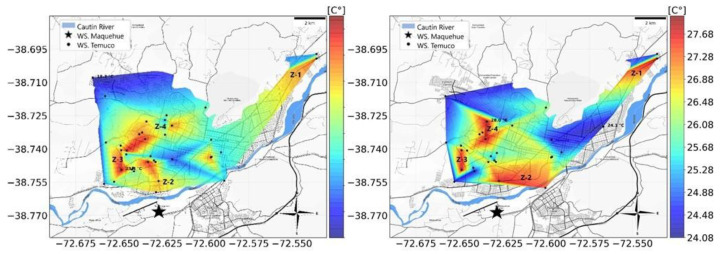
Location of the UHI in Temuco on 27 November 2017 at 9:00 p.m. (**left**) and 19 January 2018 at 8:00 p.m. (**right**) to determine the circuit to carry out the measurements with mobile transects.

**Figure 2 sensors-25-01251-f002:**
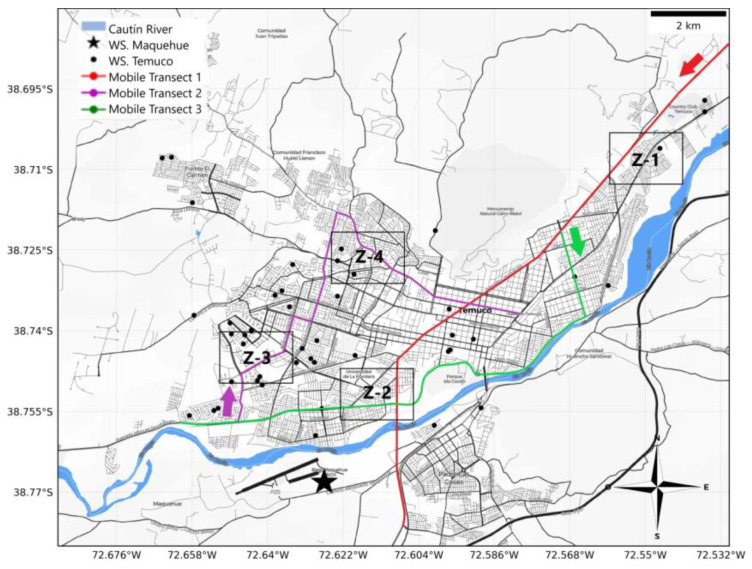
Route to capture data with the mobile transect method (red for transect 1, magenta for transect 2, and green lines for transect 3) and fixed stations distributed in different points of the city (black points).

**Figure 3 sensors-25-01251-f003:**
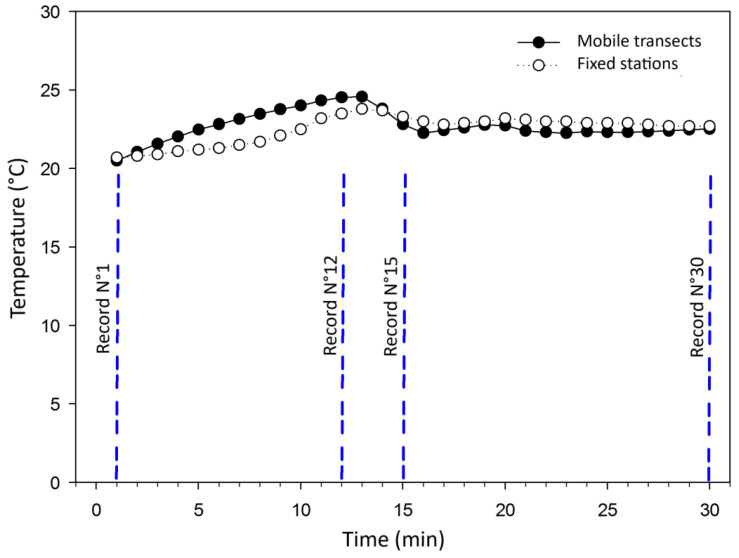
Comparison of temperatures recorded in Transect 1 and UHI simulations based on fixed stations for the time period corresponding to 20:00 h and 21:00 h.

**Figure 4 sensors-25-01251-f004:**
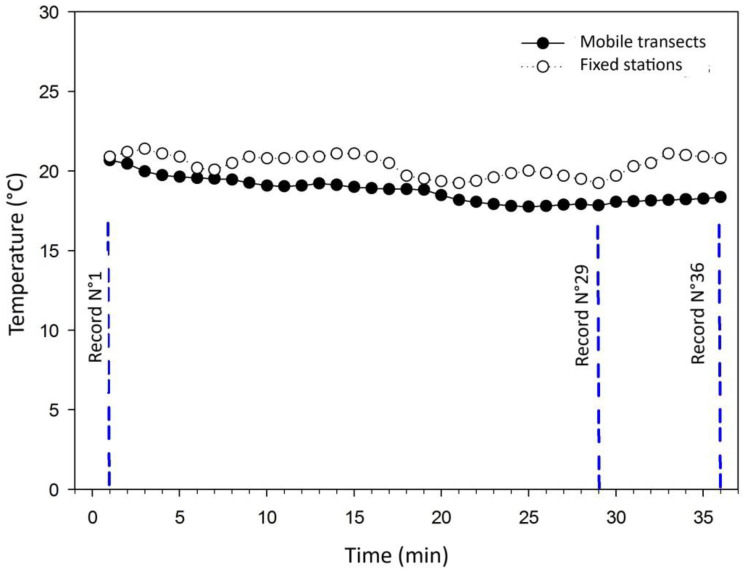
Comparison of temperatures recorded in Transect 2 and UHI simulations based on fixed stations for the time period corresponding to 20:00 h and 21:00 h.

**Figure 5 sensors-25-01251-f005:**
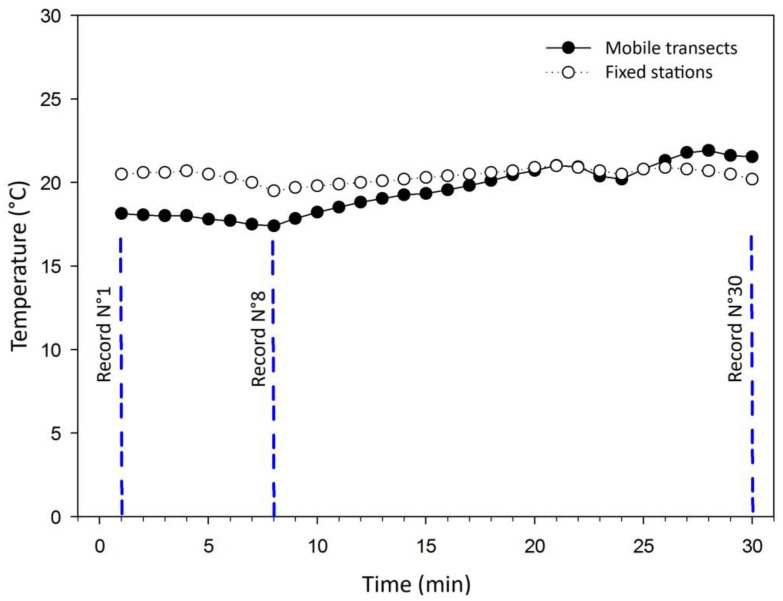
Comparison of temperatures recorded in Transect 3 and UHI simulations based on fixed stations for the time period corresponding to 20:00 h and 21:00 h.

**Figure 6 sensors-25-01251-f006:**
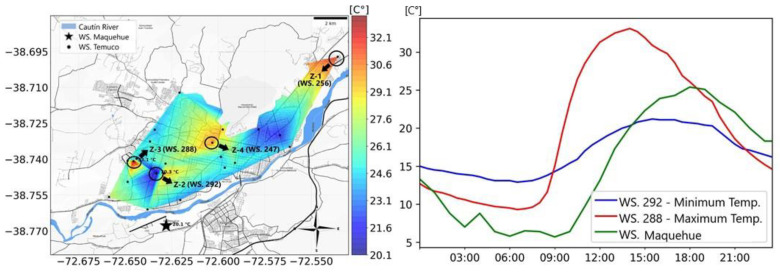
Simulation of the UHI phenomenon in Temuco using fixed station methodology (**left**) and fluctuation plot (**right**) on 4 December 2019, between sectors that have the maximum and minimum temperatures at 14:00 h.

**Figure 7 sensors-25-01251-f007:**
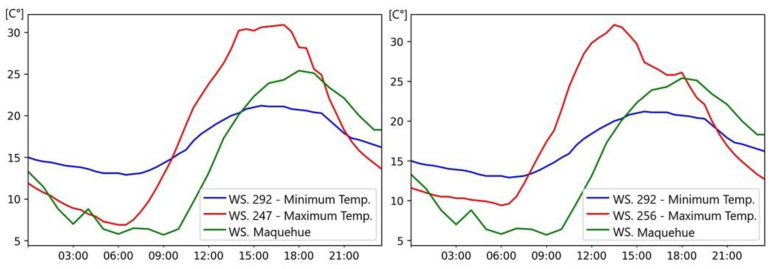
Fluctuation plot with stations in the city that record high temperatures compared to their adjacent sectors. Station 247 left plot. Zone 4 −38.733095, −72.6031539. Station 256 right plot. Zone 1 −38.697133, −72.535842.

**Figure 8 sensors-25-01251-f008:**
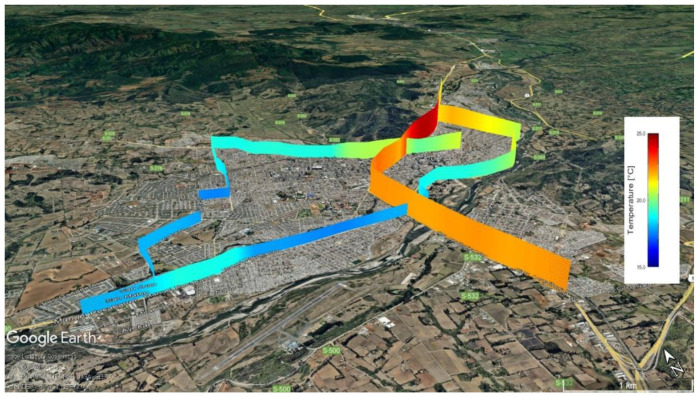
Three-dimensional representation of temperature variations across mobile transects in Temuco.

**Table 1 sensors-25-01251-t001:** Coefficient classification of Pearson’s correlation values according to Zühlke [39].

Correlation	Pearson’s Correlation Coefficient (r)	Interpretation
Perfect	r = 1	Perfect similarity, identical profiles
Very strong	0.75 ≤ r ≤ 1	Very similar, significant similarity
Medium	0.50 ≤ r ≤ 0.75	Moderate similarity
Weak	0.00 ≤ r ≤ 0.50	Little similarity
None	0.00	No correlation
Very weak	−1 ≤ r ≤ 0.00	Very large differences, opposite curves

**Table 2 sensors-25-01251-t002:** CSV file format with parameters Date-Time, ID, Temperature, Coordinates (Latitude, Longitude) processed from ReNaM data.

Date–Time	ID	T [°C]	Latitude	Longitude
2017-01-10 00:00:00	245	9.0	−38.721	−72.600
2017-01-10 00:00:00	246	9.4	−38.707	−72.665
2018-12-11 23:30:00	294	15.2	−38.740	−72.648

**Table 3 sensors-25-01251-t003:** Classification and temperature ranges associated with UHI intensity based on [43].

Classification	Intensity	Interpretation
Weak	UHI ≤ 2 °C	There is no major difference between the temperature in the city and adjacent sectors
Moderate	2 °C < UHI ≤ 4 °C	There is a slight difference between the temperature in the city and adjacent sectors not perceptible by people
Strong	4 °C < UHI ≤ 6 °C	The difference in temperature between the city and adjacent sectors is moderately perceptible by people
Very strong	6 °C < UHI ≤ 8 °C	There is a big difference in temperature between the city and adjacent sectors very perceptible by people
Extremely strong	8 °C < UHI	The difference in temperature between the city and adjacent sectors is extreme and dangerous for people

## Data Availability

Martinez-Soto, Aner (2022): Urban Heat Island Temuco. figshare. Dataset. https://doi.org/10.6084/m9.figshare.20186816.v1.

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
