# Peer review of "Heat on the Move: Contrasting Mobile and Fixed Insights into Temuco’s Urban Heat Islands"

_sensors, 2025, doi:10.3390/s25041251_

Round 1
Reviewer 1 Report
Comments and Suggestions for Authors
This subject of this manuscript focused on the phenomenon that the UHI effect significantly increased energy demand for cooling in urban areas. It is really an interesting subject.
In this manuscript, the author developed a framework for understanding UHI dynamics in diverse climatic regions and could support urban planning strategies to mitigate its impacts on quality of life.
In this work, the author used two different method including fixed stations and mobile transects to make further analysis.
Results can provide a comprehensive understanding of the phenomenon and its implications for urban planning and public health.
From the author’s view, the mobile transects can supply the spatial gradients of UHI intensity and capture urban-to-rural transitions. And the fixed stations can provide continuous and high frequency data to get the temporal analysis of diurnal and seasonal UHI patterns. Some minor questions about this point as the following.
According Table 3, the author defined a series threshold to make classification according reference[44]. Is it effective for this study region?
As Fig 3-5 shows, there shows a good consistency between two methodologies at 20:00h and 21:00h. Is it similar during the other time period?
As the author pointed in Line 320. There is abundant vegetation and most of the infrastructures are isolated houses and allows for rapid air circulation and reduces the temperature. Could you please add more details of this region for this point?
Author Response
We sincerely thank the reviewers for their time and thoughtful comments, which have greatly contributed to improving the quality and clarity of our manuscript. Their valuable insights have allowed us to refine our analysis, address key methodological considerations, and enhance the overall presentation of our findings. Below, we provide detailed responses to each comment and outline the corresponding revisions made to the manuscript.
Comment 1: According Table 3, the author defined a series threshold to make classification according reference [44]. Is it effective for this study region?
Response 1: Thank you for your observation. The thresholds in Table 3 are derived from reference [44], which is based on data from Spain and its surrounding regions. While these thresholds are broadly applicable, we acknowledge that they may not be fully tailored to the specific characteristics of our study region. This highlights the need for further research to establish locally-specific thresholds for more accurate classification and temperature assessment associated with the UHI phenomenon. We have added a note on this limitation to the discussion section of the manuscript. See page 13, lines 403-413.
Comment 2: As Fig 3-5 shows, there shows a good consistency between two methodologies at 20:00h and 21:00h. Is it similar during the other time period?
Response 2: As noted by prior research, the optimal timeframe for assessing UHI-related temperatures is between 20:00 and 21:00, as this period allows for the dissipation of solar radiation accumulated during the day, leading to more reliable temperature measurements [1], [2], [3], [4]. Measurements at other times can exhibit greater variability due to residual solar effects [5], [6]. This rationale has been clarified in the discussion section, complementing its previous mention in the methodology section. See page 13, lines 414-424.
Comment 3: As the author pointed in Line 320. There is abundant vegetation and most of the infrastructures are isolated houses and allows for rapid air circulation and reduces the temperature. Could you please add more details of this region for this point?
Response 3: The region under study is located in a suburban area of the city that has recently experienced an increase in residential construction. However, as this area was historically considered the outskirts of the city, it retains a high percentage of green spaces and vegetation. Additionally, the flat terrain facilitates consistent air circulation, contributing to temperature moderation. This description has been added to Section 3.2 of the manuscript. See page 10, lines 332-340.

Reviewer 2 Report
Comments and Suggestions for Authors
This study compares mobile transects to fixed stations to analyze 12 atmospheric UHIs
I think the study is interesting. I would like to see some information on mobile versus fixed station sensor heights. It seems to me that fixed stations may be about 2 meters height while mobile sensors could be 1+ meter in height. Please make this comparison and comment if you think it is a factor. We know that the ground temperature is usually much hotter than the air temperature. However, it would be interesting to know how temperature falls off with height.
It would be useful to see perhaps a table in the conclusion that provided a comparison with fixed versus mobile measurement issues. I think that the conclusion could be improved and I think a comparison table would be helpful.
Lastly, UHI also have anthropogenic waste heat. It would be helpful to comment on that, what issues might there be in the measurement areas for mobile vs fixed data.
Author Response
We sincerely thank the reviewers for their time and thoughtful comments, which have greatly contributed to improving the quality and clarity of our manuscript. Their valuable insights have allowed us to refine our analysis, address key methodological considerations, and enhance the overall presentation of our findings. Below, we provide detailed responses to each comment and outline the corresponding revisions made to the manuscript.
Comment 1: I think the study is interesting. I would like to see some information on mobile versus fixed station sensor heights. It seems to me that fixed stations may be about 2 meters height while mobile sensors could be 1+ meter in height. Please make this comparison and comment if you think it is a factor. We know that the ground temperature is usually much hotter than the air temperature. However, it would be interesting to know how temperature falls off with height.
Response 1: Thank you for raising this important point. The height of measurement equipment is indeed a critical factor in temperature studies, as temperature can vary significantly with height, especially near the ground where the influence of surface heat is more pronounced [7], [8], [9], [10]. To ensure consistency and reduce variability, all measurements—both from fixed stations and mobile transects—were conducted at a standardized height of 1.5 meters above the ground. This detail has been added to the methodology section. See page 8, lines 256-266.
Comment 2: It would be useful to see perhaps a table in the conclusion that provided a comparison with fixed versus mobile measurement issues. I think that the conclusion could be improved and I think a comparison table would be helpful.
Response 2: Fixed stations excel in long-term data collection and controlled conditions, which enhance accuracy and temporal resolution[11], [12], [13]. Mobile transects, while lacking in long-term resolution and requiring frequent calibration, offer superior spatial coverage and the ability to capture temperature gradients across different areas [14], [15], [16]. Although we have opted not to include a table in the conclusions to maintain the journal’s format, we have added a detailed comparative discussion to the text. If the editor approves, a comparative table is available for integration into the manuscript. See page 14, lines 478-487.
Comment 3: Lastly, UHI also have anthropogenic waste heat. It would be helpful to comment on that, what issues might there be in the measurement areas for mobile vs fixed data.
Response 3: We have expanded the discussion to address the potential biases introduced by anthropogenic heat sources in fixed station data, such as localized emissions from traffic and infrastructure [17], [18], [19]. Additionally, we discuss the transient influences affecting mobile transects, such as varying vehicular emissions and industrial activities, which may introduce noise into the measurements[20], [21]. This addition underscores the strengths and limitations of each approach in capturing spatial and temporal UHI dynamics influenced by anthropogenic activities. The revised discussion is included in the discussion section. See page 13, lines 425-434
